# Prognostic Impact of Sarcopenia in Patients with Advanced Prostate Carcinoma: A Systematic Review

**DOI:** 10.3390/jcm12010057

**Published:** 2022-12-21

**Authors:** Pedro de Pablos-Rodríguez, Tasmania del Pino-Sedeño, Diego Infante-Ventura, Aythami de Armas-Castellano, Miguel Ramírez Backhaus, Juan Francisco Loro Ferrer, Pedro de Pablos-Velasco, Antonio Rueda-Domínguez, María M. Trujillo-Martín

**Affiliations:** 1Department of Urology, Instituto Valenciano de Oncología (IVO), 46009 Valencia, Spain; 2Doctoral School of University of Las Palmas de Gran Canaria (ULPGC), 35001 Las Palmas de Gran Canaria, Spain; 3Canary Islands Health Research Institute Foundation (FIISC), 38320 Santa Cruz de Tenerife, Spain; 4Evaluation Unit of the Canary Islands Health Service (SESCS), 38109 Santa Cruz de Tenerife, Spain; 5Network for Research on Chronicity, Primary Care, and Health Promotion (RICAPPS), 38109 Santa Cruz de Tenerife, Spain; 6Department of Clinical Sciences, University of Las Palmas de Gran Canaria (ULPGC), 35001 Las Palmas de Gran Canaria, Spain; 7Department of Endocrinology and Nutrition, University Hospital of Gran Canaria Doctor Negrín, 35012 Las Palmas de Gran Canaria, Spain; 8Research Institute of Biomedical and Health Sciences (IUIBS), University of Las Palmas de Gran Canaria (ULPGC), 35001 Las Palmas de Gran Canaria, Spain; 9Medical Oncology Intercenter Unit, Regional and Virgen de la Victoria University Hospitals, IBIMA, 29590 Malaga, Spain; 10Research Network on Health Services in Chronic Diseases (REDISSEC), Carlos III Health Institute, 28029 Madrid, Spain

**Keywords:** prostatic neoplasms, sarcopenia, prognosis, survival, systematic review, meta-analysis

## Abstract

Prostate cancer (PCa) is the second most common cancer in men and the fifth leading cause of death from cancer. The possibility of sarcopenia being a prognostic factor in advanced PCa patients has recently become a subject of interest. The aim of the present study was to evaluate the prognostic value of sarcopenia in advanced prostate carcinoma. A systematic review was conducted in Medline, EMBASE, and Web of Science (March, 2021). The quality of studies was assessed using the Quality in Prognosis Studies tool. Meta-analyses for overall, cancer-specific, and progression-free survival were performed. Nine studies (n = 1659) were included. Sarcopenia was borderline associated with a shorter overall survival (HR = 1.20, 95% CI: 1.01, 1.44, P = 0.04, I^2^ = 43%) but was significantly associated with progression-free survival (HR = 1.61, 95% CI: 1.26, 2.06, P < 0.01; k = 3; n = 588). Available evidence supports sarcopenia as an important prognostic factor of progression-free survival in patients with advanced PCa. However, sarcopenia has a weak association with a shorter overall survival. The evidence on the role of sarcopenia in prostate-cancer-specific survival is insufficient and supports the need for further research. Patient summary: The literature was reviewed to determine whether the loss of muscle mass (sarcopenia) affects the survival in patients with advanced PCa. Patients with advanced PCa and sarcopenia were found to have a shorter progression-free survival (the length of time during and after treatment of a cancer that the patient lives with the disease but it does not get worse), but sarcopenia did not have much influence on the overall survival and cancer-specific survival (the length of time from either the date of diagnosis or the start of treatment to the date of death due to the cancer).

## 1. Introduction

Prostate cancer (PCa) is a global health problem, with approximately 1.4 million cases diagnosed worldwide each year [1]. It is the second most common type of cancer in men, after lung cancer [2], and the fifth leading cause of death from cancer [3]. The mean age of PCa onset is sixty-five and the majority of cases are diagnosed from that age onward [3].

There is an inherent decrease in serum testosterone concentrations as age increases, directly affecting the development of muscle mass and fat mass. Consequently, the ageing process accelerates the development of sarcopenia [4]. Sarcopenia is a progressive and musculoskeletal disease linked to the chronological age of the person [5], characterized by the loss of muscle mass and its associated function, which is associated with an increased likelihood of adverse outcomes including falls, fractures, physical disability, and mortality [5]. Specifically, sarcopenia is probable when low muscle strength is detected. A sarcopenia diagnosis is confirmed by the presence of low muscle quantity or quality. When low muscle strength, low muscle quantity/quality, and low physical performance are all detected, sarcopenia is considered severe [5].

Furthermore, one of the therapeutic approaches in the different PCa scenarios may involve androgen deprivation therapy (ADT), which produces adverse effects such as changes in body composition and physical function [6]. Therefore, the age-related loss of muscle and fat mass and, therefore, sarcopenia is accentuated in these patients by the effect of the therapy [6].

To date, sarcopenia has been identified as a poor prognostic factor for disease progression and mortality in patients with ovarian cancer [7], breast cancer [8], lung cancer [9], or colorectal cancer [10], among others. In the case of PCa, although different prognostic factors for disease progression and mortality have been established including Gleason score, clinical stage, prostate-specific antigen (PSA), presence of visceral or liver metastases, and number of metastatic sites [11], there is a lack of a clear conclusion or consensus about the prognosis role of sarcopenia. Determining this association may potentially inform the design of specific and tailored strategies to improve the prognosis of PCa patients and the effectiveness of the first-line treatments.

Therefore, the aims of the study were to identify, critically assess, and synthesize the available scientific evidence on the impact of sarcopenia on disease progression and mortality in patients with advanced PCa.

## 2. Evidence Acquisition

A systematic review was developed following the Cochrane Prognosis Methods Group [12] and the reporting followed the Preferred Reporting Items for Systematic Reviews and Meta-Analyses (PRISMA) statement [13]. The protocol is registered in the PROSPERO database (reference number CRD42021248645).

### 2.1. Information Sources and Search Strategy

Medline (using the Ovid platform), EMBASE, and Web of Science (WOS) databases were searched (March 26, 2021). The search strategy was initially developed in Medline including both controlled vocabulary and text-word terms related to sarcopenia and prostate neoplasms and then adapted for each of the other databases. Searches were restricted to the English and Spanish languages and no time limits were imposed. The search strategy is available in Appendix A (Appendix A). The reference lists of all relevant papers were examined to identify possible additional studies meeting selection criteria.

### 2.2. Study Selection Process

Studies were eligible for inclusion if they fulfilled the following criteria:

(a) Type of study: Any longitudinal observational study (e.g., cohort studies, case–control studies, or database linkage studies) and secondary analyses of experimental studies (randomized or non-randomized) investigating the prognosis significance of sarcopenia in patients with PCa for predicting mortality or disease progression were included. For an experimental study to be eligible, it must have used either the control group alone or the entire study sample adjusted for the intervention. Cross-sectional studies, case series, or case studies and systematic or narrative reviews were excluded.

(b) Population: Studies that evaluated men aged sixty and older diagnosed with advanced PCa were included. Patients were considered advanced if they had metastatic, hormone-sensitive, or castration-resistant PCa (nodal, bone, and/or visceral) defined as cTxNxM1.

Studies including only a subset of the participants relevant to the review question, such as studies including patients with other types of cancer in addition to patients with PCa, were included as long as the results for patients meeting the inclusion criteria were reported separately or they accounted for more than 80% of the target population.

Studies conducted with healthy volunteers or animals were excluded.

(c) Index prognostic factor: the presence of sarcopenia defined as progressive and the generalized loss of skeletal muscle mass and function assessed by magnetic resonance imaging (MRI), computed tomography scan (muscle area or muscle volume or skeletal muscle index—SMI), dual-energy X-ray absorptiometry (SMI), or bioelectrical impedance analysis—BIA (SMI).

(d) Comparator: absence of sarcopenia.

(e) Outcome measures: studies had to report on overall survival (OS), cancer-specific survival, overall response rate to cancer treatment, progression-free survival (PFS), complications of cancer, or health-related quality of life (HRQL).

(f) Timing: sarcopenia measurement had to be performed during or after diagnosis.

No study based on the duration of follow-up was excluded.

(g) Setting: studies conducted in primary or secondary healthcare were included.

(h) Language: only studies published in English or Spanish were included.

### 2.3. Study Selection Process

The study selection process was conducted by two reviewers as follows: first, the reviewers screened independently and in duplicate the titles and abstracts of all the retrieved citations; secondly, the reviewers, again independently and in duplicate, read and evaluated for inclusion the full text articles that appeared to fulfil the pre-determined selection criteria. The reviewers compared and discussed results in both phases and consulted a third reviewer in case of doubt and discrepancy.

### 2.4. Data Collection Process

A data extraction form (in Excel format) was prepared by the authors, pilot-tested on three studies before the start of the data extraction process, and refined accordingly. Two reviewers independently and in duplicate extracted the following data from the included studies: identification of the article (author, year of publication, country, and funding), design and methodology (objective, number of centers, and duration of follow-up), population and their demographics (e.g., sample size, age, cancer grade/stage, and metastases), sarcopenia (definition, measurement method, timing, and cut-off point), and outcomes and the results of the study (means, event counts, hazard ratio—HR, or odds ratio—OR, with special attention to the variability in the results presented (standard deviation, variance, *p*-values, etc.)). HRs and ORs were extracted from univariate and multivariate analyses. A third reviewer subsequently verified the extracted data.

### 2.5. Risk of Bias Assessment

Again, two reviewers independently and in duplicate assessed the potential risk of bias in the studies included using the Quality in Prognosis Studies (QUIPS) tool [14]. Each of the six domains used by QUIPS includes multiple items that are judged separately. Based on the ratings of the items, a conclusive judgment of the risk of bias within each domain was made and expressed on a three-grade scale (low, moderate, or high risk of bias). In the systematic review here, the overall risk of bias was considered low if up to one domain was rated as at moderate risk of bias. If one or more study domains were rated as at high risk or if three or more were rated as at moderate risk, the study was then classified as at high risk of bias. All studies in between were classified as having moderate risk of bias [15].

The inter-rater agreement using the weighted Kappa and percent agreement was assessed. Discrepancies of judgments between the reviewers were discussed and, in case no consensus could be achieved, a third reviewer was consulted. The QUIPS-files are available upon request from the authors.

### 2.6. Assessment of Publication Bias

According to the recommendations of the Cochrane Collaboration [16], publication bias was examined by constructing a funnel plot and computing the Egger test, with the significance level set at 0.05, using metafunnel and metabias commands in STATA version 16 (StataCorp LLC, College Station, TX, USA).

### 2.7. Analysis and Synthesis of Results

A meta-analysis was performed for outcomes reported by two or more studies. The meta-analysis and forest plot for the sarcopenia rate were calculated using the metaprop command in STATA version 16. The hazard ratio (HR) and the corresponding 95% CI for OS, cancer-specific survival, and PFS were pooled with an indirect variance estimation in meta-analyses using the statistical program Review Manager (RevMan, version 5.4.1. Copenhagen: The Nordic Cochrane Center, The Cochrane Collaboration, 2020), and results were displayed in forest plots. Heterogeneity was assessed using the I^2^ statistic. When there was heterogeneity (I^2^ ≥ 50% or P < 0.1), meta-analyses were performed using a random-effects model. A sensitivity analysis was conducted by omitting each study individually to determine the stability of the overall estimate of the effect. The effects of disease stage (castration-sensitive or castration-resistant PCa) and treatment type (androgen deprivation therapy plus chemotherapy or alone; or chemotherapy) were explored using subgroup analyses. The nature of the data reported for age, presence of metastases, and sarcopenia stage did not allow them to be grouped for the analysis. Meta-regression was also not possible, due to the small number of studies evaluated.

### 2.8. Certainty of Evidence Assessment

An assessment of the certainty of evidence per outcome was performed based on the Grading of Recommendations Assessment, Development, and Evaluation (GRADE) approach. Certainty could be rated down considering five domains: risk of bias, inconsistency, indirectness, imprecision, and publication bias; or rated up considering three domains: large effect, dose−response gradient, and plausible confounding [17]. Evidence profiles were built and the overall certainty of evidence was rated from very low (little confidence in the estimate; the true prognosis is likely to be substantially different from the estimate) to high (very confident that the true prognosis is close to that of the estimate).

## 3. Evidence Synthesis

The results of the literature search and study selection process are shown in Figure 1. Out of a total of 861 initially identified references after eliminating duplicates, 164 potentially relevant articles were selected for full text assessment. Nine studies were finally eligible for inclusion according to the pre-established selection criteria [18,19,20,21,22,23,24,25,26], and eight of them were selected for quantitative synthesis [18,19,20,21,22,24,25,26]. All selected studies were published in English between 2015 and 2021. The list of studies excluded at the full-text level and the reasons for exclusion are provided in Appendix A.

### 3.1. Description of Included Studies

The main characteristics of the selected studies are summarized in Table 1. Seven studies were retrospective medical record reviews [18,21,22,23,24,25,26] and two were retrospective cohorts [19,20]; studies were conducted in South Korea [20,21,22], Japan [20,24], the United Kingdom [19], France [18], Spain [23], and Austria [26].

Across the nine studies, 1659 men were recruited. The largest study consisted of 411 men [22], whereas the smallest study had only 59 men [23].

The mean age of the patients was 69.77 years (SD: 1.85) ranging from 61 to 78 years of age. Five studies focused on patients with castration-resistant PCa [21,22,24,25,26], two studies on patients with metastatic castration-resistant PCa [18,19], and two others on patients with metastatic PCa [20,23]. The overall prevalence of sarcopenia was 61% (95% CI: 46– 76%; I^2^ = 97.07%, *P* > 0.01) (Appendix A). However, three of the included studies did not report the number of participants with sarcopenia; the data of these studies could not be included in the analysis [19,21,23].

The most commonly used method for sarcopenia screening of participants was measuring SMI using a CT scan at L3 [18,19,20,21,22,25,26]. Nonetheless, one study used the L3-psoas muscle index [24] and another study [23] used the European Working Group on Sarcopenia in Older People criteria [5]. The criteria used to define sarcopenia are shown in Table 1. Finally, sarcopenia measurement was performed at the time of disease diagnosis, PCa diagnosis [23,26], or castration-resistant PCa diagnosis [21,22] in four studies; before starting treatments in three studies [19,20,25]; at tumor assessment in one study [18]. One study did not report on the time point of the sarcopenia measurement [24].

Seven studies considered OS as the clinical outcome [18,19,20,22,23,24,25,26], two considered cancer-specific survival [20,21], and four considered PFS [18,21,25,26]. None reported data on overall response rates, complications of cancer, or HRQL. The mean duration of the reported follow-up was twenty-nine weeks [18,20,21,22,23,26].

### 3.2. Risk of Bias in Included Studies

Risk of bias was considered high in two of the nine included studies [19,26] and moderate in one study [20]. Study attrition and prognostic factor measurement bias were suspected in one study due to the failure to account for confounding concerns in the exclusion criteria [26]. A probable study-confounding bias was identified in three studies due to the partial information on the measurement and analysis of all important confounders [19] or the method and setting of confounding measurement [20,26]. In general, in the domain of statistical analyses and reporting bias, analysis intentions were not available, or not reported in sufficient detail to enable an assessment. The detailed judgements for each of the risk of bias domain criteria are shown in Table 2.

The percent agreement was 82% and the inter-rater agreement was moderate (Kappa = 0.56).

### 3.3. Synthesis of Results

Out of the nine included studies, eight that included 1600 patients remained for quantitative analysis [18,19,20,21,22,24,25,26]. The study excluded from quantitative analyses did not determine a cut-off for defining sarcopenia but instead analyzed muscle mass as a continuous variable [23]. Results of all meta-analyses and subgroup and sensitivity analysis are available in Appendix A.

The quality of evidence ranged from high to very low. Appendix A provides the evidence profile for sarcopenia-related outcomes.

No evidence of publication bias was detected through visual assessment (Appendix A) or from the result of Egger’s regression test for each pooled outcome, except in univariate analysis of PFS (P = 0.02) (see Appendix A).

#### 3.3.1. Overall Survival

The meta-analyses of the univariate [18,19,20,22,24,25,26] and multivariable data [19,20,22,24,25] on the influence of sarcopenia on OS is shown in Figure 2. The pooled results of univariate data showed that PCa patients with sarcopenia had a significantly higher risk of all-cause mortality (fixed effects, HR = 1.44, 95% CI: 1.23, 1.67, P < 0.01, I^2^ = 0%; k = 7; n = 1081) versus participants without sarcopenia. In the multivariate data meta-analysis (fixed effects), there was a borderline significant association between sarcopenia and OS (fixed effects, HR = 1.20, 95% CI: 1.01, 1.44, P = 0.04, I^2^ = 43%; k = 5; n = 831).

The subgroup and sensitivity analysis showed no statistically significant changes in the overall outcome estimate, as shown in Appendix A.

#### 3.3.2. Cancer-Specific Survival

Only two studies [20,21] reported on cancer-specific survival. In the meta-analysis of the univariate data, there was no significant association between sarcopenia and cancer-specific survival (random effects, HR = 1.98, 95% CI: 0.80, 4.90, P = 0.14; I^2^ = 74%; k = 2; n = 479) (Figure 3). However, the only study that reported a multivariate model [20] showed that sarcopenia was significantly associated with shorter cancer-specific survival (HR = 2.18, 95% CI: 1.07, 7.32, P = 0.04; n = 197).

Subgroup analysis of univariate data indicated a statistically significant association in patients with hormone-sensitive PCa (HR = 3.48, 95% CI: 1.43, 8.47, P = 0.05; n = 197).

#### 3.3.3. Progression-Free Survival

Four studies provided data on PFS [18,21,25,26]. The pooled analysis demonstrated an association between sarcopenia and shorter PFS, and this association existed in both univariate (HR = 1.56, 95% CI: 1.29, 1.88, P < 0.01; I^2^ = 0%; k = 4; n = 818) and multivariate analyses (HR = 1.61, 95% CI: 1.26, 2.06, P < 0.01; k = 3; n = 588) (Figure 4).

The subgroup analysis of univariate data suggested no significant effect of treatment on the association between sarcopenia and PFS.

## 4. Discussion

The findings reported here support sarcopenia as being an important prognostic factor of PFS in patients with advanced PCa. Additionally, a less clear association between sarcopenia and cancer-specific survival or OS was also found.

As PFS is a surrogate outcome of cancer-specific survival and OS, the fact that the results of the present review have been conclusive only for this variable and not for cancer-specific survival and OS may be related to the short duration of follow-up in the included studies (mean: 29 weeks). It is likely that a longer follow-up could demonstrate a clearer positive association between sarcopenia and survival/mortality variables.

As with the results obtained for other types of cancer [7,8,9,10], two recent meta-analyses identified sarcopenia as a poor prognostic factor for disease progression in PCa [27,28]; however, none have focused on sarcopenia as a prognostic factor for advanced PCa. In addition, the effect of sarcopenia on overall survival was assessed in both studies but not on cancer-specific survival and PFS. Finally, our subgroup analyses and the assessment of the certainty of evidence were not performed in these previous studies.

Sarcopenia prevalence in patients with PCa estimated by SMI in the included studies ranged widely from 50.36 [22] to 82.80% [26]. This may be due to the use of different cut-offs for sarcopenia diagnosis (45.2–55 cm^2^/m^2^), including the use of an obesity-specific SMI cut-off. As the cut-off used to define sarcopenia directly influences the outcome of associations made between SMI and prognosis in cancer patients, it is necessary for a consensus to be reached on this.

The prevalence of sarcopenia in patients with PCa is markedly high (61%) as compared to patients affected by other types of cancer (38.6%) [29]. This higher prevalence can be explained by two factors: first, the advanced mean age of the sample (69.77 years), and secondly, because a significant percentage of the patients were under ADT.

The main result obtained in the present review is the association between sarcopenia and PFS, which could be explained by the worse treatment response that patients with sarcopenia experience [29].

It is easier to explain the relationship obtained between sarcopenia and OS (weak association but statistically significant on multivariate analysis). This association has been shown in other solid tumors [30,31,32,33]. Sarcopenia assumed decreased functional reserves. The poor functional reserves are associated with the frailty phenotype. The close relationship between sarcopenia and frailty functional syndrome is probably the main reason behind the findings here concerning OS.

Moreover, in patients requiring surgery, surgical procedures that minimize the risk of worsening sarcopenia should be prioritized. In this sense, it is widely known that outpatient surgery or minimally invasive surgery involving fewer days of hospitalization compared to conventional surgery can help reduce the risk of malnutrition and, consequently, the worsening of sarcopenia that hospitalization entails [34].

When ADT is required, intermittent ADT may be an alternative to reduce the impact of hypogonadism on muscle. From the oncological point of view, this strategy has shown non-inferiority with respect to continuous ADT [35,36]. In fact, the European urology guidelines endorse intermittence as an ADT treatment option in a selected profile of patients [37]. In the same way that intermittence can attenuate the impact on bone mass [38], it could perhaps attenuate its effect on muscle mass.

Current European Urology Guidelines only mention sarcopenia as a consequence of androgenic treatment. However, as sarcopenia could be an unfavorable prognostic factor that can be worsened by PCa treatment, it should be systematically screened and, if detected, patients should receive personalized treatment [6,37]. On the other hand, the diagnosis of sarcopenia should be accompanied by other measures to reduce the impact of hypogonadism on the muscle. Thus, preliminary studies have shown that physical exercise programs can improve sarcopenia in patients with PCa [39], even in the absence of testosterone [40]. Improvements in sarcopenia have also been obtained in patients with PCa supplemented with high doses of vitamin D [41,42]. In addition, different studies are currently being conducted to assess the effect of protein and creatinine supplementation, but the results have not been published yet [43,44].

The main limitation of the present review is that the evidence comes exclusively from retrospective studies, a design characterized by poor control over the exposure factor, covariates, and potential confounders and bias. In addition to the short follow-up periods in the studies included in our review, another important limitation is, due to the lack of consensus on the definition of sarcopenia, the diversity of cut-off points used by the considered studies for assessing sarcopenia. Moreover, subgroup and meta-regression analyses to explore the effect of important variables such as age, presence of metastases, and sarcopenia stage on the magnitude of association could not be performed. Finally, another potential limitation of this review is the possibility that some studies have not been included, because they are not written in English or Spanish or because they are not indexed in the consulted databases. Despite all these limitations, the present study benefits from rigorous methods following the fundamental principles of transparency and replicability, a comprehensive search, a peer selection, data extraction and risk of bias assessment, a quantitative synthesis of results with the exploration of important potential sources of heterogeneity, and an assessment of the certainty of evidence on the basis of a structured and explicit approach.

For sarcopenia diagnosis, the following cut-off points are arbitrary at this time; the development of validated cut-off points depends on normative data and their predictive value for hard endpoints, which is a high priority for research studies.

In addition to new studies with a longer-term follow-up on the effects of sarcopenia on advanced PCa progression, future lines of research should be related to the analysis of the impact of different measures aimed at eliminating or attenuating sarcopenia and their effect on the evolution of advanced PCa, such as nutrition [45], physical exercise [46], or whether intermittence is an ADT treatment option in patients with advanced PCa. Other aspects in which the evaluation of the role of sarcopenia could be relevant are related to the decision to start treatment or not, or whether or not to combine treatment, as well as the decision on the type of treatment that best suits a muscle state. In this respect, several studies, some of which are included in the meta-analysis here, support the view that patients with sarcopenia suffer greater toxicity and have worse tolerance to chemotherapy [24,25,47,48]. The proposal of these studies is to use sarcopenia as a factor to decide to treat patients with a new generation of antiandrogens rather than with chemotherapy. The fact that the new antiandrogens also lead to a reduction in lean body mass is known [49,50], but whether or not this is related to decreased survival or disease-free time has not been established [25].

In conclusion, the available evidence supports the view that sarcopenia is an important prognostic factor of PFS in patients with advanced prostate carcinoma. However, sarcopenia has a weak association with a shorter OS. Finally, the available evidence on the role of sarcopenia in cancer-specific survival is insufficient and, as such, precludes drawing definitive conclusions and, furthermore, supports the need for further research efforts.

## 5. Take Home Message

Sarcopenia is an important prognostic factor of PFS in patients with advanced PCa.

Sarcopenia has a weak association with a shorter overall survival.

There is a lack of evidence on the role of sarcopenia in cancer-specific survival.

## Figures and Tables

**Figure 1 jcm-12-00057-f001:**
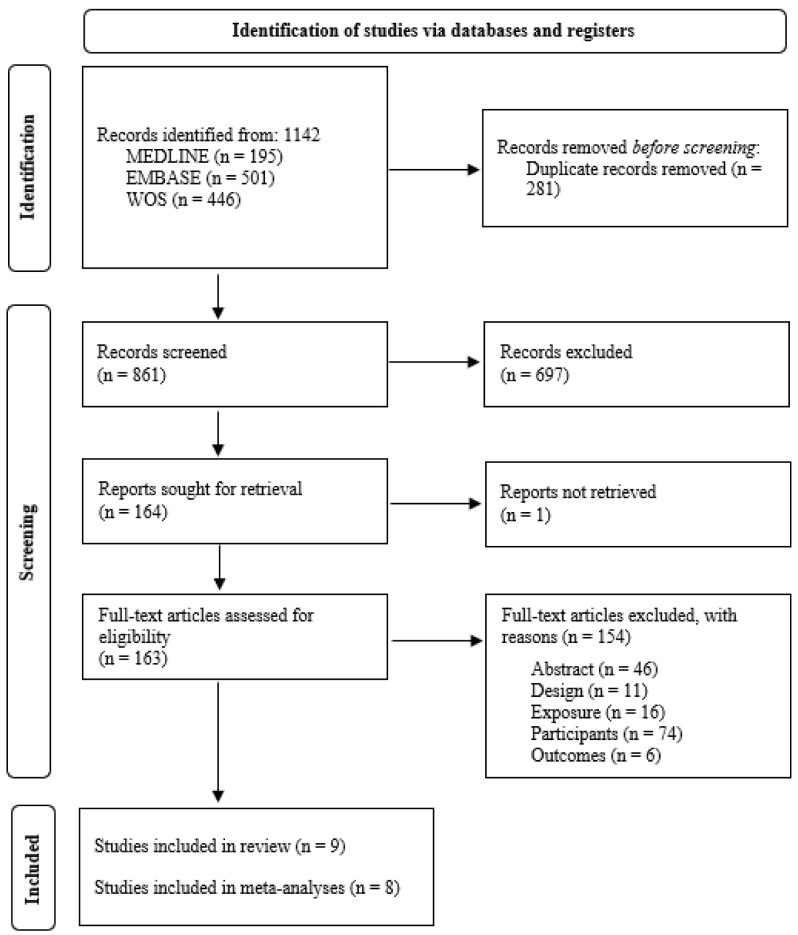
PRISMA flow chart detailing the screening process.

**Figure 2 jcm-12-00057-f002:**
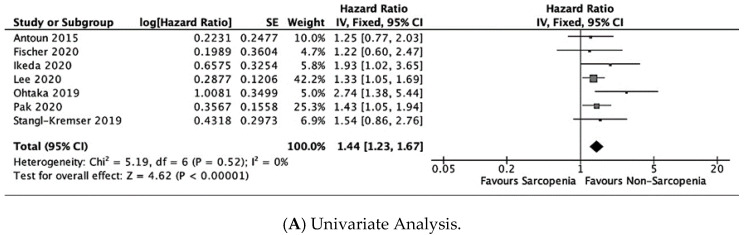
Forest plots for overall survival [10,11,12,13,14,15,16,17,18,19,20,22,24,25,26].

**Figure 3 jcm-12-00057-f003:**
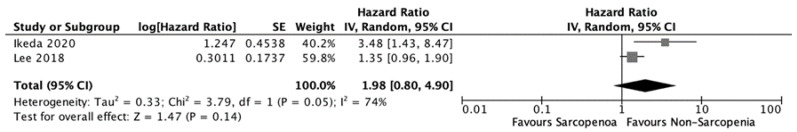
Forest plot for cancer-specific survival—Univariate Analysis [20,21].

**Figure 4 jcm-12-00057-f004:**
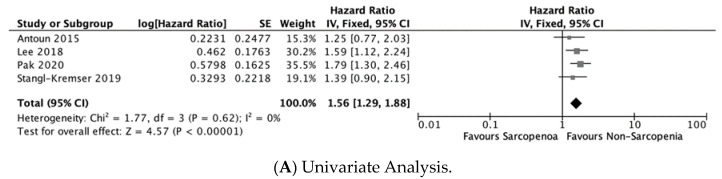
Forest plots for progression-free survival [18,21,25,26].

**Table 1 jcm-12-00057-t001:** Characteristics of included studies.

First Author, Year (Country)	Design	N	Patient					Sarcopenia			Outcomes	Follow-Up Time (Months)
			Age ^a^	Inclusion Criteria	Exclusion Criteria	Metastases (N)	Cancer Treatment (%)	Definition	Prevalence (%)	Diagnosis Method		
Antoun, 2015 [18] (France)	RRR	127	69 (63–74) *	Metastatic CRPCa	NR	Visceral: 15	1. Enzalutamide + prednisolone: 62 2. Abiraterone + prednisolone: 24	SMI < 43 cm^2^/m^2^ (BMI < 25 kg/m^2^) SMI < 53 cm^2^/m^2^ (BMI > 25 kg/m^2^)	66.14	At L3 by CT scan	OS	16 (95% CI: 12–19)
											PFS	Sarcopenia: 4 (95% CI: 3–6) Non-sarcopenia: 5 (95% CI: 3–6)
Fischer, 2020 [19] (UK)	RCS	90	69 (NR) *	Starting treatment (enzalutamide or abiraterone) for metastatic CRPCa	No imaging data	Bone: 15 Bone and lymph node: 17 Lymph node: 10 Visceral: 1	Previous ADT: 100% 1. Enzalutamide + prednisolone: 69% 2. Abiraterone + prednisolone: 31%	SMI < 52.4 cm^2^/m^2^	NR	At L3 by CT scan	OS	NR
Ikeda, 2020 [20] (Japan)	RCS	197	73 (66.0–78.0) *	1. Metastatic hormone-sensitive PCa 2. No previous treatments	Missing clinical or imaging data at diagnosis	Bone: 127 Distant lymph node: 29 Other locations: 9	Previous ADT:100% 1. Docetaxel: 39.6% 2. Cabazitaxel: 6.7% 3. Enzalutamide: 30.6% 4. Abiraterone: 34.3% 5. Other treatment: 17.1%	SMI < 33 cm^2^/m^2^ (BMI < 25 kg/m^2^) SMI < 53 cm^2^/m^2^ (BMI >25 kg/m^2^)	82,74	At L3 by CT scan	OS	Sarcopenia: 72 (IQR: 50–84) Non-sarcopenia: NR (IQR: 52-NR)
											CSS	Sarcopenia: 77 (IQR: 62-NR) Non-sarcopenia: NR (IQR: 75-NR)
Lee, 2018 [21] (Republic of Korea)	RRR	282	67.0 (61.0–72.0) *	CRPCa progression	1. Incomplete clinical data 2. Loss to follow-up 3. Unknown cause of death	Bone: 155 Lymph node: 118 Visceral: 10	Previous ADT: 100% 1. Docetaxel + prednisolone: NR 2. Enzalutamide + prednisolone: NR 3. Abiraterone + prednisolone: NR	SMI < 52.4	NR	At L3 by CT scan)	CSS	15
											PFS	3.7
Lee, 2020 [22] (Republic of Korea)	RRR	411	70 (65–76) *	CRPCa progression	1. Insufficient imaging data 2. Lost to follow-up 3. Unknown cause of death	Bone: 344 Lymph node: 199 Visceral:70	Previous ADT:100% 1. Docetaxel + prednisolone: NR 2. Cabazitaxel + prednisolone: NR 3. Enzalutamide + prednisolone: NR 4. Abiraterone + prednisolone: NR	SMI < 45.2 cm^2^/m^2^ SMA < 32.4 HU	50.36	At L3 by CT scan	OS	Sarcopenia: 19 Non-sarcopenia: 24
Muñoz-Rodríguez, 2021 [23] (Spain)	RRR	59	72.74 (12.25)	Metastatic onset PCa + first-line ADT	No imaging data	Bone: 52 Retroperitoneal lymphadenopathy: 30 Visceral: 6	1. ADT: 100%	European Working Group on Sarcopenia in Older People criteria [5]	NR	CT scan	OS	32.3 (95% CI: 17.1–47.16)
Ohtaka, 2019 [24] (Japan)	RRR	77	70 (65–76) *	CRPCa + docetaxel chemotherapy	NR	Bone: 55 Lymph node: 34 Visceral: 12	1. Previous ADT + docetaxel+ prednisolone: 100%	Psoas muscle index < 5.7 cm^2^/m^2^	33.77	At L3-psoas muscle by CT scan	OS	16.41 (IQR: 10.85–25.97)
Pak, 2020 [25] (Republic of Korea)	RRR	230	68.3 (9.1)	CRPCa + first-line therapy	1. Insufficient imaging data before starting first-line treatment 2. Patients treated for <2 months 3. Patients followed-up for <6 months	Bone: 196 Lymph node: 122 Solid organ: 28	Previous ADT: 100% 1. Docetaxel + prednisolone: 7.0% 2. Cabazitaxel + prednisolone: 24.3% 3. Enzalutamide + prednisolone: 10.0% 4. Abiraterone + prednisolone: 13.0% 5. Other treatment + prednisolone: 2.1%	SMI < 50 cm^2^/m^2^	51.30	At L3 by CT scan	OS	Sarcopenia: 16.9 Non-sarcopenia: 24.1
											PFS	Sarcopenia: 9.1 Non-sarcopenia: 14.9
Stangl-Kremser, 2019 [26] (Austria)	RRR	186	68,8 (64.6–75.0) *	CRPCa + chemohormonal therapy	1. Insufficient imaging data 2. Lost to follow-up	Bone: 146 Distant lymph node: 65 Liver: 16 Visceral (No liver): 19	1. Docetaxel + prednisolone: 100	SMI < 55 cm^2^/m^2^ (men)	82.80	At L3 by CT scan	OS	26.2 (IQR 13.7–42.4)
											PFS	7.8 (IQR: 4.4–16.3)

^a^ Mean (SD) or median (IQR); * as reported. ADT: androgen deprivation therapy; BMI: body mass index; CRPCa: Castration-resistant prostate cancer; CSS: Cancer-specific survival; CT: Computerized Tomography; DSF: Progression-free survival; HU: Hounsfield Units; CI: confidence interval; IQR: interquartile range; NR: not reported; OS: Overall survival; PCa: Prostate cancer; RCS: Retrospective cohort study; RRR: Retrospective record review; SD: standard deviation; SMA: Skeletal Muscle Attenuation; SMI: Skeletal muscle index; UK: United Kingdom.

**Table 2 jcm-12-00057-t002:** Risk of bias assessment.

Study	Study Participation	Study Attrition	Prognostic Factor Measurement	Outcome Measurement	Study Confounding	Statistical Analysis and Reporting	Overall Risk of Bias
Antoun [18]	Low	Low	Low	Low	Low	Moderate	Low
Fischer [19]	Low	Low	Low	Low	High	High	High
Ikeda [20]	Low	Low	Low	Low	Moderate	Moderate	Moderate
Lee [21]	Low	Low	Low	Low	Low	Moderate	Low
Lee [22]	Low	Low	Low	Low	Low	Moderate	Low
Muñoz-Rodríguez [23]	Low	Low	Low	Low	Low	Moderate	Low
Ohtaka [24]	Low	Low	Low	Low	Low	Moderate	Low
Pak [25]	Low	Low	Low	Low	Low	Moderate	Low
Stangl-Kremser [26]	Low	High	High	Low	Moderate	Moderate	High

Low: low risk of bias; Moderate: moderate risk of bias; High: high risk of bias.

## Data Availability

The data presented in this study is available in the supplementary material.

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
