# Peer review of "Prognostic Impact of Sarcopenia in Patients with Advanced Prostate Carcinoma: A Systematic Review"

_jcm, 2022, doi:10.3390/jcm12010057_

Round 1

Reviewer 1 Report

Thank you very much for submitting this study to the journal. Although the manuscript has a significant body of data, I have several concerns regarding the manuscript.

Major points

-First of all, the end date of the search needed to be updated (passed over 1.5 years). Therefore, the study could benefit from a more current literature search after a protocol revision in PROSPERO. In this form, the manuscript has the same date interval as a previous meta-analysis (DOI: 10.1016/j.urolonc.2021.08.009). Furthermore, the manuscript only added one study to quantitative analyses compared to an earlier meta-analysis with a broader scoop (DOI: 10.1038/s41391-021-00442-0). Several articles were published on this topic last year, although the search strategy did not allow the authors to include these studies (DOI: 10.3390/cancers13246345, DOI: 10.1155/2022/9242243).

-The authors should discuss the added benefit of the present meta-analysis to the literature compared to previous meta-analyses. The previous two meta-analyses were not cited in the manuscript.

-The authors consider adding subgroup analyses for overall survival to the results according to disease stage (castration-sensitive and castration-resistant) and treatment type (androgen receptor pathway inhibitor and chemotherapy). It could be more clinically relevant for the prognostic role of sarcopenia rather than conducting subgroup analyses for cancer-specific survival.

Minor points

-The numbers do not add up in the PRISMA diagram and the text. The diagram suggested the inclusion of seven articles in the quantitative analyses, while it was written as eight studies on page 5, line 197. Also, regarding the `about the records removed before screening` part of the flow diagram, I could not understand the logic behind adding reasons for study exclusion, while no studies were excluded due to these reasons (Zero records for both `Records marked as ineligible by automation tools` and `Records removed for other reasons`).

-The authors could consider adding a column to the table-1 demonstrating the cut-off (AUC, median, etc.) selection in the particular articles.

-Although sometimes used interchangeably, the disease-free survival term is preferred for patients treated with curative intent, while progression-free survival was mostly used for the advanced stage disease. Please, consider changing disease-free survival to progression-free survival in the manuscript.

Author Response

Dear reviewer,

We sincerely appreciate all your comments and suggestions to our manuscript. Below we provide the responses to each comment and the specific changes included in the manuscript.

Comments and Suggestions for Authors

Thank you very much for submitting this study to the journal. Although the manuscript has a significant body of data, I have several concerns regarding the manuscript.

Major points

-First of all, the end date of the search needed to be updated (passed over 1.5 years). Therefore, the study could benefit from a more current literature search after a protocol revision in PROSPERO. In this form, the manuscript has the same date interval as a previous meta-analysis (DOI: 10.1016/j.urolonc.2021.08.009). Furthermore, the manuscript only added one study to quantitative analyses compared to an earlier meta-analysis with a broader scoop (DOI: 10.1038/s41391-021-00442-0). Several articles were published on this topic last year, although the search strategy did not allow the authors to include these studies (DOI: 10.3390/cancers13246345, DOI: 10.1155/2022/9242243).

Response: Although the time elapsed since searching the electronic databases does not seem too long to us considering the laboriousness of a systematic review process, we agree with the reviewer that the article could benefit from an update of the search. In any case, we cannot carry out this task within the 10-day response period that we have been given. On the other hand, we believe that our work, in its current form, has value and contributes to scientific knowledge since it would be the first published meta-analyses carried out with this scope and objectives. Please, see the response to the next comment for differences and benefits regarding recent meta-analysis (DOI: 10.1016/j.urolonc.2021.08.009) (DOI: 10.1038/s41391-021-00442-0).

-The authors should discuss the added benefit of the present meta-analysis to the literature compared to previous meta-analyses. The previous two meta-analyses were not cited in the manuscript.

Response: The two meta-analyses mentioned by the reviewer were published after our search date in the electronic databases, therefore, we were not aware of these studies so they were not cited in our manuscript. We are very grateful for this contribution. Although both had evaluated the prognostic value of sarcopenia in patient with PCa, none had linked sarcopenia with the prognosis of advanced PCa. In addition, both studies only focused on the effect of sarcopenia on overall survival, leaving behind other important variables for the patient and clinical decision making. Moreover, no subgroup analyses have been performed in these studies. Finally, an assessment of the certainty of evidence was not carried out either in these previous meta-analyses.

As recommended by the reviewer, we have expanded the paragraph dedicated to compare our meta-analyses with the previous one adding all these differences and benefits mentioned considering also the two new meta-analyses (lines 316-323). It has been leaved as follows:

Like the results obtained for other types of cancer [7–10], two recent meta-analyses identified sarcopenia as a poor prognostic factor for disease progression in PCa [27,28], however, none have focused on sarcopenia as a prognostic factor for advanced PCa. Besides, the effect of sarcopenia on overall survival is assessed in both studies but not on cancer-specific survival and PFS. Finally, no subgroup analyses nor assessment of the certainty of evidence were performed in these previous studies.

The citations of both studies have been added to the reference list as 27 and 28, and subsequent citations have been renumbered accordingly.

-The authors consider adding subgroup analyses for overall survival to the results according to disease stage (castration-sensitive and castration-resistant) and treatment type (androgen receptor pathway inhibitor and chemotherapy). It could be more clinically relevant for the prognostic role of sarcopenia rather than conducting subgroup analyses for cancer-specific survival.

Response: We have added the subgroup analyses according to disease stage and treatment type. We have reported this at different points in the manuscript:

  • In the subsection 2.7. Analysis and synthesis of results, the following sentence has been included (page 4, lines 181-184):

The effect of disease stage (castration-sensitive or castration-resistant PCa) and treatment type (androgen deprivation therapy plus chemotherapy or alone; or chemotherapy) were explored using subgroup analysis.

In addition, we have modified the adjoining sentence leaving it as follows: The nature of the data reported for age, presence of metastases and sarcopenia stage did not allow them to be grouped for the analysis. Meta-regression was also not possible due to the small number of studies evaluated.

  • In the section 3.3. Synthesis of results:

- in line 261, we have added “,subgroup”

- in lines 295-296, we have added the sentence: Subgroup analysis of univariate data indicated a statistically significant association in patients with hormone sensitive PCa (HR = 3.48, 95% CI: 1.43, 8.47, P = 0.05; n = 197).

- in lines 309-310, we have added the sentence: The subgroup analysis of univariate data suggested no significant effect of treatment on the association between sarcopenia and PFS.

3) In Discussion section, within the paragraph of limitations, we have modified a sentence leaving it as follows (lines 377-380): Moreover, subgroup and meta-regression analyses to explore the effect of important variables such as age, presence of metastases and sarcopenia stage on the magnitude of association could not be performed.

4) Finally, results of these new analyses have been included in Supplementary Table 3.

Minor points

-The numbers do not add up in the PRISMA diagram and the text. The diagram suggested the inclusion of seven articles in the quantitative analyses, while it was written as eight studies on page 5, line 197. Also, regarding the `about the records removed before screening` part of the flow diagram, I could not understand the logic behind adding reasons for study exclusion, while no studies were excluded due to these reasons (Zero records for both `Records marked as ineligible by automation tools` and `Records removed for other reasons`).

Response: we are very sorry for these mistakes. PRISMA diagram has been corrected, replacing 7 with 8 in studies included in meta-analyses and removing from the box “Records removed before screening” the reasons for study exclusion where no studies were excluded.

-The authors could consider adding a column to the table-1 demonstrating the cut-off (AUC, median, etc.) selection in the particular articles.

Response: Cut-off for sarcopenia in each study is already included in the column “Definition”.

-Although sometimes used interchangeably, the disease-free survival term is preferred for patients treated with curative intent, while progression-free survival was mostly used for the advanced stage disease. Please, consider changing disease-free survival to progression-free survival in the manuscript.

Response: We agree with the reviewer that the most appropriate term is progression-free survival. We have replaced disease-free survival with progression-free survival throughout the text. Consequently, DFS has been changed to PFS. In addition, the definition of the term that appears within bracket in the patient summary of the abstract has been changed to the definition corresponding to the new term.

Reviewer 2 Report

This study aims to evaluate the prognostic value of sarcopenia in advanced prostate carcinoma. It is very well written and scientifically sound. Therefore I only have some minor textual issues that should be addressed:

- Abstract: perhaps remove the inline headers such as "Evidence acquisition" and "Evidence Synthesis".

- Line 208: "androgens deprivation therapy"-> "androgen deprivation therapy"

- Line 211: "standard desviation" -> "standard deviation"

- Line 253: "sensitive analysis"-> "sensitivity analysis"

- Figure 4: "Figure 4. Forest plots for progression-free survivalB Multivariate Analysis." -> the last part should be a sub-header within the figure?

- References: please correct the double numbers (e.g. "1. [1]")

- The Word file with Suppl. Table 2 also contains a number of figures. These should be removed.

Author Response

Dear reviewer,

We sincerely appreciate all your comments and suggestions to our manuscript. Below we provide the responses to each comment and the specific changes included in the manuscript.

Comments and Suggestions for Authors

This study aims to evaluate the prognostic value of sarcopenia in advanced prostate carcinoma. It is very well written and scientifically sound. Therefore I only have some minor textual issues that should be addressed:

- Abstract: perhaps remove the inline headers such as "Evidence acquisition" and "Evidence Synthesis".

Response: Sorry for this mistake. We ignore the instructions for authors of the journal regarding the abstracts: structured abstracts, but without headings. We have deleted headings.

- Line 208: "androgens deprivation therapy"-> "androgen deprivation therapy"

Response: amended

- Line 211: "standard desviation" -> "standard deviation"

Response: amended

- Line 253: "sensitive analysis"-> "sensitivity analysis"

Response: amended

- Figure 4: "Figure 4. Forest plots for progression-free survivalB Multivariate Analysis." -> the last part should be a sub-header within the figure?

Response: amended

- References: please correct the double numbers (e.g. "1. [1]")

Response: amended

- The Word file with Suppl. Table 2 also contains a number of figures. These should be removed.

Response: we don't understand this comment. What number of figures are you referring to? However, thanks to this comment we have realized that we forgot to reference this supplementary table in the text. At the end of the first paragraph of Evidence Synthesis section (page 5), we have added the following sentence:

The list of studies excluded at the full-text level and the reasons for exclusion is provided in Supplementary Table 2.

Accordingly, the following supplementary tables have been renumbered.

Reviewer 3 Report

Line 81 – “fist-line” should be changed to “first-line”

Line 124 – The abbreviation “(OS)” should probably be inserted after “overall survival” here.

Line 204-205 – It seems unusual that no Sweden or United States studies were found that had data on sarcopenia in prostate cancer.  Just a suggestion, but some comment on why no Swedish or U.S. studies were found, might be interesting.

Line 206 – “Table 1. Characteristics of studies included.”  This title, of course, should be moved onto the same page as the table.

Table 1 –  The formatting of this table is currently very poor, difficult to read, and needs substantial revision.  At the very least, the columns should be expanded to use the full width of the standard page.  However, it might be better to reformat it into Landscape fashion, i.e., rotated 90 degrees on page.

Line 217-219 – This statement seems a bit problematic to the reviewer,  i.e., calling CRPC and Metastatic PCa “two different types” of PCa.  These entities are frequently part-and-parcel of the same disease state.  Did they mean metastatic CRPC versus metastatic HSPC (hormone sensitive prostate cancer) ?   Please revise or explain this statement.

Line 233-234 – Some comment on this relative short mean duration of follow up (29 weeks) should be made either here or in the Discussion section.  Does this possibly effect the overall quality of the data assessment, for instance?

Line 264 – “PCA” should be changed to “PCa” to be consistent

Line 289 – “univariable” should be changed to  “univariant”

Line 294 – The subtitle “B Multivariate Analysis” needs to be separated from the figure legend; it should be at least one line up from the legend.

Line 297-298 – The sentence here should probably be stated a little differently.  I would suggest, e.g., that,  while there is “a less clear association” between sarcopenia and CSS and OS, there is some suggestion of an association existing.

Line 327 – the phrase “,may be an alternative” is redundant and should be removed.

Suggest that the three paragraphs (lines 332-343) be condensed into a single paragraph.

Author Response

Dear reviewer,

We sincerely appreciate all your comments and suggestions to our manuscript. Below we provide the responses to each comment and the specific changes included in the manuscript.

Comments and Suggestions for Authors

Line 81 – “fist-line” should be changed to “first-line”

Response: done

Line 124 – The abbreviation “(OS)” should probably be inserted after “overall survival” here.

Response: done

Line 204-205 – It seems unusual that no Sweden or United States studies were found that had data on sarcopenia in prostate cancer.  Just a suggestion, but some comment on why no Swedish or U.S. studies were found, might be interesting.

Response: We agree with the reviewer that it is striking that two countries with large databases on PCa do not have any contribution in this matter, however we don't have an explanation for that. If you have any suggestions or anything that you might find interesting to add to the manuscript, we would gladly do it.

Line 206 – “Table 1. Characteristics of studies included.”  This title, of course, should be moved onto the same page as the table.

Response: amended.

Table 1 –The formatting of this table is currently very poor, difficult to read, and needs substantial revision.  At the very least, the columns should be expanded to use the full width of the standard page.  However, it might be better to reformat it into Landscape fashion, i.e., rotated 90 degrees on page.

Response: we have changed the orientation of the table to landscape,which has allowed the columns to be expanded. We think it's much better now.

Line 217-219 – This statement seems a bit problematic to the reviewer,  i.e., calling CRPC and Metastatic PCa “two different types” of PCa.  These entities are frequently part-and-parcel of the same disease state.  Did they mean metastatic CRPC versus metastatic HSPC (hormone sensitive prostate cancer) ?   Please revise or explain this statement.

Response: We agree with the reviewer that calling CRPC and Metastatic PCa “two different types” of PCa is not correct and the sentence is confusing. Consequently, we have deleted from the sentence (lines 224-227 in the revised document) the text “Two different types of PCa were examined, with”. As we meant type of patients, the sentence has been rewritten as follows:

Five studies focused on patients with castration-resistant PCa [21,22,24–26], two studies on patients with metastatic castration-resistant PCa [18,19] and two others on patients with metastatic PCa [20,23].

Line 233-234 – Some comment on this relative short mean duration of follow up (29 weeks) should be made either here or in the Discussion section.  Does this possibly effect the overall quality of the data assessment, for instance?

Response: We fully agree with the reviewer that something must be said about the duration of follow up. We are sorry we overlooked it. We have included the following paragraph in the Discussion section (lines 316-320):

Since PFS is a surrogate outcome of cancer-specific survival and OS, the fact that the results of the present review have been conclusive only for this variable and not for cancer-specific survival and OS may be related to the short duration of follow-up in the included studies (mean: 29 weeks). It is likely that longer follow-up could demonstrate a clearer positive association between sarcopenia and survival/mortality variables.

In addition, we have pointed out the short duration of follow-up in the included studies as a limitation of our meta-analyses (lines 374-375).

Finally, to point out the evident need for more studies in this line of research and with longer follow-up periods, we have added the following text at the beginning of the penultimate paragraph of the Discussion section (lines 392-393):

In addition to new studies with longer-term follow-up on the effects of sarcopenia on advanced PCa progression,

Line 264 – “PCA” should be changed to “PCa” to be consistent

Response: done.

Line 289 – “univariable” should be changed to  “univariant”

Response: done.

Line 294 – The subtitle “B Multivariate Analysis” needs to be separated from the figure legend; it should be at least one line up from the legend.

Response: amended.

Line 297-298 – The sentence here should probably be stated a little differently.  I would suggest, e.g., that, while there is “a less clear association” between sarcopenia and CSS and OS, there is some suggestion of an association existing.

Response: we accepted the reviewer's suggestion. The modified sentence is as follows: Additionally, a less clear association between sarcopenia and cancer specific survival or OS was also found.

Line 327 – the phrase “,may be an alternative” is redundant and should be removed.

Response: done

Suggest that the three paragraphs (lines 332-343) be condensed into a single paragraph.

Response: we fully agree with the reviewer. The three paragraphs have been merged.

Round 2

Reviewer 1 Report

I would like to thank the authors for their kind response. As stated by the authors, updating the search would be hard during the limited time for revision. Therefore, I think that it would be better for the journal to give an extended revision period to update the search.

Other than this point, my other points were revised accordingly.